# Supervised treatment in outpatients for schizophrenia plus (STOPS+): protocol for a cluster randomised trial of a community-based intervention to improve treatment adherence and reduce the treatment gap for schizophrenia in Pakistan

Thomas Andrew Shepherd [1], Zia Ul-Haq,[2] Mian Ul-Haq,[3] Muhammad Firaz Khan,[3] Adil Afridi,[3] Lisa Dikomitis,[1,4] Michelle E Robinson,[1] Martyn Lewis,[1] Atif Rahman,[5] Krysia Dziedzic,[1] Umaima Saeed,[2] Naila Riaz Awan,[3] Christian Mallen,[1,6] Saeed Farooq[1,6]

**Correspondence to**
Dr Saeed Farooq;
s.farooq@keele.ac.uk

## ABSTRACT

**Introduction** There is a significant treatment gap, with only a few community-based services for people with schizophrenia in low-income and middle-income countries. Poor treatment adherence in schizophrenia is associated with poorer health outcomes, suicide attempts and death. We previously reported the effectiveness of supervised treatment in outpatients for schizophrenia (STOPS) for improving treatment adherence in patients with schizophrenia. However, STOPS was evaluated in a tertiary care setting with no primary care involvement, limiting its generalisability to the wider at-risk population. We aim to evaluate the effectiveness of STOPS+ in scaling up the primary care treatment of schizophrenia to a real-world setting.

**Methods and analysis** The effectiveness of the STOPS+ intervention in improving the level of functioning and medication adherence in patients with schizophrenia in Pakistan will be evaluated using a cluster randomised controlled trial design. We aim to recruit 526 participants from 24 primary healthcare centres randomly allocated in 1:1 ratio to STOPS+ intervention and enhanced treatment as usual arms. Participants will be followed-up for 12 months postrecruitment. The sample size is estimated for two outcomes (1) the primary clinical outcome is level of functioning, measured using the Global Assessment of Functioning scale and (2) the primary process outcome is adherence to treatment regimen measured using a validated measure. An intention-to-treat approach will be used for the primary analysis.

**Ethics and dissemination** Ethical approval has been obtained from Keele University Ethical Review Panel (ref: MH-190017) and Khyber Medical University Ethical Review Board (ref: DIR-KMU-EB/ST/000648). The results of the STOPS+ trial will be reported in peer-reviewed journals and academic conferences and disseminated to local stakeholders and policymakers.

### Strengths and limitations of this study

► This study aims to evaluate an approach that can be used for scaling up treatment for schizophrenia in resource poor settings and specifically addresses the gap in schizophrenia treatment.
► To the best of our knowledge, this is first study that uses a directly observed treatment short-course approach translated from an infectious disorder, that is, tuberculosis to a non-communicable disorder for scaling up the treatment for schizophrenia.
► The intervention involves primary care in treatment of schizophrenia in community, which is rarely done in low-income and middle-income country.
► The trial is powered two primary outcomes (1) a clinical outcome measure—Global Assessment of Functioning and (2) a process outcome measure—adherence to medication.
► The measurement of treatment adherence is based on a subjective measure, that is, reporting from patients and carers that cannot be objectively validated.

**Trial registration number** ISRCTN93243890.

## INTRODUCTION

Schizophrenia is one of the leading causes of years lived with disability, especially in low-income and middle-income countries (LMICs).[1] Existing evidence-based interventions for schizophrenia have been poorly implemented and most LMICs have large treatment gap for schizophrenia.[2] The treatment gap represents the disparity between the

true prevalence of a disorder and the treated proportion of individuals diagnosed.[3] Pakistan has one of the largest treatment gaps for schizophrenia, at an estimated 96%, this is contrast to an estimated median of 32.2% globally and 69% in LMICs.[4][5] This treatment gap is attributed to a number of factors including poor treatment adherence with average non-treatment adherence rates of about 50% in schizophrenia,[4] level of disability, a lack of primary care involvement and little to no access to treatment.[5] This is a particularly important factor in rural areas, where healthcare resources are often diminished in comparison to urban centres.[6] While the cost of antipsychotic medication in Pakistan is relatively low, economic and logistical factors remain significant barriers to medication access[7] and the purchase of antipsychotic medication is often at the expense of other forms of essential out-of-pocket healthcare or even food.[8]

Reducing the treatment gap for schizophrenia in LMICs is critical. Medication non-adherence in patients with schizophrenia is associated with 2.8 times increased risk in hospital admission.[9] The risk of death in patients who receive even minimal treatment is significantly reduced compared with patients receiving no treatment at all.[10] Furthermore, untreated psychosis can at least double the costs of managing chronic comorbid physical health conditions, leading to an increased stress on an already overburdened healthcare system.[11]

We previously reported a randomised controlled trial (RCT) of supervised treatment in outpatients for schizophrenia (STOPS) in Peshawar, Pakistan.[12] STOPS was developed using the Medical Research Council framework for complex interventions[13] and was modelled on the directly observed treatment short-course (DOTS) intervention to improve treatment adherence for people with tuberculosis (TB). The DOTS approach involves a TB worker or a family member directly supervising the patient taking all medication and recording this.[14] DOTS is one of the most successful public health interventions and has been shown to be as highly effective in the community as it is in specialist clinics, which in addition to its accessibility and low cost, which make it a highly appropriate intervention model for LMICs. Using the similar approach, STOPS intervention involved a family member of a person with schizophrenia observing and recording that medication had been taken. The family member is trained for this purpose. A previous evaluation of STOPS in a tertiary care setting showed that the STOPS significantly improved treatment adherence and functioning as well reducing patient symptoms.[11]

The original STOPS intervention was evaluated in a tertiary care setting by mental health professionals, with no primary care involvement, limiting its scaling up and generalisability. Through a programme of qualitative work with key stakeholders in Pakistan, including patients, carers, healthcare professionals, researchers and local traditional healers, we adapted the STOPS model to STOPS+ with the aim of scaling up the intervention and involving primary care in the management of schizophrenia.

Here, we outline the protocol for evaluating the STOPS+ in a real-world setting using a cluster RCT design.

## METHODS AND ANALYSIS
### Objectives
#### Primary objective
The primary objective is to evaluate the clinical effectiveness of the STOPS+ intervention in improving medication adherence and functioning in patients with schizophrenia compared with enhanced treatment as usual (ETAU) in primary care in Pakistan.

#### Secondary objectives
1. To evaluate the cost-effectiveness of the STOPS+ intervention compared with ETAU.
2. To assess the effectiveness of STOPS+ in reducing family/caregiver burden and stigma in the community.
3. To evaluate whether STOPS+ leads to physical health improvement in patients with schizophrenia.
4. To investigate the implementation of STOPS+ in a primary care setting and the acceptability of STOPS+ for service users and healthcare providers, assessing the impact of STOPS+ in the wider healthcare system.

### Study setting
The intervention will be implemented and evaluated in the Peshawar district of Khyber Pakhtunkhwa (KP). KP is one of four provinces in Pakistan located in the North West region with an estimated population of 20.7 million. Health expenditure in Pakistan is about 1% of the gross domestic product with no separate provision for mental health services. The primary healthcare (PHC) facilities in KP are provided by a network of basic health units, civil dispensaries and rural health centres. Typically, each PHC facility consists of a primary care physician, a multipurpose PHC technician (MT) and a female health worker.[15] Training PHC providers to manage mental health conditions and the incorporation of mental health services into primary care are both envisaged in the current health policy but have not yet been fully realised.[5]

### Trial design
A cluster RCT will be used to investigate the efficacy of the STOPS+ intervention compared with ETAU in people living with schizophrenia. The cluster RCT design will help to evaluate the implementation of STOPS+ at the healthcare facility level and to minimise the contamination of intervention and control arms.

### The control: ETAU
Typically, in Pakistan, treatment for schizophrenia consists of accessing hospital-based psychiatric services with little to no involvement of PHC and a limited provision of medication from the hospital pharmacy. ETAU provided in the control clusters will include the treatment received by patients' routine healthcare setting, which includes treatment provided by a psychiatrist in the outpatient clinics of the psychiatry department of the local hospital

and brief counselling about the treatment and outcome of the disorder. Patients may be admitted to an inpatient unit at the hospital. This will be enhanced in the control cluster PHCs with training provided to PHC physicians in WHO Mental Health Gap Action Programme (MhGAP) (https://www.who.int/mental_health/mhgap/en/t) in addition to the regular and reliable provision of psychotropic drugs at the PHC. The patients will be informed that the antipsychotic medication is available in the PHC centre and they can access the centre for supply of medication. The PHC physician will have the option to liaise with the treating psychiatrist, as provided in the MhGAP guidelines.

### STOPS+ intervention
The STOPS+ intervention will consist of the ETAU plus the following components.

### Supervision by a trained family member for dispensing and administering medication
A family member who is living with the patient for a minimum of 6 months prior to recruitment to the trial will be nominated by the patient. The family member will dispense medication, observe that is has been taken correctly and record this information on a simple sheet of paper designed and used in the previous STOPS trial.[9] Family members will be reminded to administer the medication by an automated text message system, which will also request a response once the medication has been taken correctly. The text message reminders will be sent everyday for the first 3 months after starting the treatment. If the family member does not have access to a mobile phone, a handset and training in using it will be provided.

A multipurpose technician (MPT) from each PHC centre will provide training to family members for the storage and administration of the medication, recording when the medication has been taken correctly and the importance of maintaining the family dynamic.

### Treatment for schizophrenia at PHC level supervised by mental health professionals
Ongoing management and monitoring of the patient's schizophrenia will take place at their PHC centre by the PHC physician, under the supervision of the treating psychiatrist with the help of the mobile technology platform smartphone for monitoring, assessment and remote treatment (SMART). SMART will provide two-way communication between mental healthcare professionals. One month's supply of medication will be made available, via their PHC, to the family member. The automated messaging system will also remind the family member about collecting the following month's supply.

### Monitoring the availability of essential psychotropic medication and their side effects
A kit to monitor the supply of medication will be used to monitor the availability and supply of psychotropic medication. This will consist of (a) a cupboard dedicated specifically for the storage of psychotropic medication, (b) a tracking sheet used to monitor the stock levels and (c) a simple checklist for monitoring side effects of antipsychotics.

### Assessment of barriers and facilitators of implementation of the intervention
The use of theory to inform the development of behaviour change interventions is strongly advocated by experts in the field. We have designed the process of implementation using theory developed for implementation and tested previously in UK Primary Care Practice.[16 17] We will evaluate the barriers and facilitators to implementation of the intervention using qualitative methods. We will hold at least four focus groups (with patients and carers, MPTs, traditional and faith healers and primary care clinicians) in which we will explore the barrier and facilitators to supervising the treatment of schizophrenia in the community, modification of the STOPS programme to STOPS+ for use in the community and the present status of schizophrenia care in the community

### Patient and public involvement
STOPS+ is underpinned by culturally appropriate and context-bespoke patient and public community engagement and involvement at every stage of the project (intervention development, RCT and process evaluation). The STOPS+ intervention will be refined with input from the key stakeholder groups including patients, carers, health professionals, local councillors, community leaders such as religious leaders and community experts and policymakers. Small group meetings using local Jirgas (Jirga is traditional, local participatory and decision-making body which is well established in Pathan culture and brings together local tribal, ethnic and religious leaders) will help in community engagement.

### Primary outcome measure
In view of the pragmatic nature of the trial, we will use two distinct primary outcomes:
1. The primary clinical outcome will be the level of functioning, measured using the Global Assessment of Functioning (GAF) scale.[18] GAF is a commonly used tool in schizophrenia, particularly for the pragmatic trials.[19]
2. The primary process outcome will be adherence to treatment regimen. The adherence to treatment will be measured using a questionnaire adapted from Herz et al.[20] This measure for assessing treatment adherence was successfully used in our previous STOPS study.[9]

### Secondary outcomes
The details of secondary outcomes and the standardised instruments for measuring these are given in table 1. These instruments will measure secondary outcomes such as physical health outcomes, family/caregivers burden and internalised stigma.

**Table 1** Description of outcome measures and instruments

| Domains measured | Measure |
|---|---|
| Sociodemographics | Age |
| | Gender |
| | Years in education |
| | Marital status |
| | Work status |
| | Mini-Neuropsychiatric Interview for diagnosis of schizophrenia[21] |
| Physical health | Body mass index |
| | Blood pressure |
| | Waist circumference |
| Primary outcomes | |
| Level of functioning | Global Assessment of Functioning[18] |
| Level of adherence | Treatment adherence rating scale[20] |
| Secondary outcomes | |
| Mental state and psychiatric symptoms | Brief Psychiatric Rating Scale[40] |
| Caregivers' burden | Family Burden Scale[41] |
| Perceived stigma | Internalised Stigma of Mental Illness[42] |
| Side effects of antipsychotic medication | Glasgow Antipsychotic Side-Effects Scale[43] |
| Cost of care | Client Service Receipt Inventory[44] |
| Quality of life and cost-effectiveness | EuroQol (EQ5D)[45] |
| Illness severity, improvement and response to treatment | Clinical Global Impression Scale[46] |
| Drug use | DAST Drug Screening Tool[47] |
| Depression | Patient Health Questionnaire[48] |
| Suicide Ideation | Suicide Behaviours Questionnaire Revised[49] |

DAST, Drug Abuse Screening Test; EQ5D, EuroQol 5 Dimension.

## Eligibility criteria

Patients will be eligible for inclusion if they

a. Have a diagnosis of schizophrenia (F20.9) or schizoaffective disorder (F25.9) based on the International Classification of Disease 10 criteria assessed using the Mini-International Neuropsychiatric Interview.[21]

b. Are aged between 17 and 65 years.

c. Do not meet the criteria for remission as defined by the Remission in Schizophrenia Working Group.[22]

d. Have capacity and are able to give informed consent.

e. Have a family member who is also willing to participate in the trial and supervise the treatment after training by the research team.

Exclusion criteria will be kept to a minimum to ensure that the study can examine the real-world implementation of the STOPS+. The patients who can not participate in the study due to one of following reasons will be excluded (1) a serious or unstable medical illness, (2) evidence of learning disability, (3) severe drug dependence requiring treatment and/or detoxification and (4) pregnant or breast-feeding.

## Duration of treatment and follow-up

Both groups will be followed up for 12 months after recruitment in the study. The follow-up using face-to-face assessments will be carried out by trained research assistants in a health facility which is not part of participating PHCs to preserve the blindness of clusters. All of the outcomes will be measured at baseline, at 6 months and at 12 months by research assistants.

## Participant identification

Participants will be identified from two possible sources (1) patients living in the area of the participating PHCs and presenting at psychiatry outpatient services of the hospitals in the area and (2) potentially eligible patients identified by the PHC workers during community engagement or attending the PHCs. Eligible participants will be informed about the trial orally by the PHC staff and written information will be provided in the local language (Urdu). Those interested in taking part will be asked to give written informed consent. For participants who cannot read and write, witnessed oral consent and a thumbprint in lieu of a signature will be used. The witness will not be a member of the research team. The procedure has been used in our previous trials in the area.[23]

## Randomisation and allocation concealment

Randomisation will occur at the PHC level, where PHCs will be allocated at a ratio of 1:1 to either deliver the STOPS+ intervention or ETAU. Twenty-four PHCs will be recruited to the study, 12 will be randomised to deliver ETAU and 12 will be randomised to deliver the STOPS+ programme. PHC randomisation will be stratified by urban/rural setting. Randomisation will be carried out by an independent statistician at the Khyber Medical University (KMU), Peshawar by using a remote computer-generated random sequence. Randomisation and will be based on the list of primary care centres in Peshawar, provided by the health department of KP. The urban/rural classification is determined as per the Peshawar district council. It is not possible to blind study participants from their treatment arm allocation. However, the assessment team that conducts the baseline and follow-up assessments (6 and 12 months) will be masked to the treatment arm of the cluster. The outcome measures will be administered by trained research assessment teams (outcome assessors), who are independent of the intervention team and the team involved in consenting and screening. The outcome assessment will not be carried out at any participating PHC. The assessment teams will be told that they

are evaluating two interventions and that there is genuine equipoise about which one is better. Study participants will be instructed to not discuss the PHC they are attending or the treatment details. During all assessments, the primary outcome measures (treatment adherence scale and GAF) will be completed first to minimise the risk of bias in the event of unmasking and, if it occurs, the point of unmasking will be recorded. Sensitivity analyses will be carried out to assess the effect of unmasking on the primary outcomes. The data linking each PHC with treatment allocation status is kept separate from the outcome dataset until the time of the final analysis. The trial statistician will be blind to allocation status.

### Screening, data collection and follow-up

After identification of potentially eligible patients, they will first be orally informed about the trial by one of the health workers in the local Pushto language (MT in the community or a health professional in one of the health facilities mentioned above). They will be asked whether they agree that a member of the research team will provide them with further information about the research. If permission is given, a research assistant will meet with the patient and will request informed consent. The informed consent for participation in the study will be taken within 1 week of the first contact. Participants will be free to decline to participate or withdraw at any time without their routine healthcare being affected. Before taking part in any interviews, oral and written information about the study and its purpose will be provided to respondents in the local language.

After obtaining informed consent, the research assistant will decide whether the patient meets the inclusion criteria and have a family member who is willing to supervise the treatment as specified in the STOPS+ procedures. If the patient is not eligible from the screening assessment, the reason for ineligibility will be explained to the patient and they will be signposted for further advice and to continue to receive treatment at their routine health facility. Eligible patients will be provided with a further participant information sheet about the trial. Additionally, for those in the intervention arm, the nominated relative will receive a participant information sheet about the trial and about the tasks required in supervising the treatment If nominated family member is available with the patient, this information will be provided at the same time. If the family member is not available, patients and the family member will be asked to visit the health facility again within 1 week. The patient and the relative will each be given the opportunity to ask any questions about the trial. The information sheets are provided in Urdu. For illiterate patients, the research assistants will explain the same information in local Pushto language or in Urdu. Following the informed consent, the baseline data will be collected.

### Sample size calculation

A mean difference of 6 on the primary clinical outcome measure (GAF) with a pooled study SD of 15 was observed in our previous trial in the same setting.[12] A recent indicated minimum clinically important difference for the GAF in schizophrenia patients has been given as a mean difference of 4.[24] A total sample size of 526, that is, 263 patients in each arm (recruited across 24 PHCs, with an average cluster size of 22 patients) has over 90% power to detect a mean difference of 6 and 85%–90% power to detect a more modest mean difference of 4 in the GAF given a two-tailed significance level of 0.05 and based on the following parameter assumptions: (1) baseline–outcome correlation of 0.5, repeated measures correlation of 0.7[25] and loss to follow-up of 20% (patient-level parameters) and (2) intracluster correlation coefficient of 0.01[26] and coefficient of variation in cluster sizes of 0.65 (PHC-level parameters).[27] The primary endpoint for the GAF is the 'average' GAF across 6 and 12 months follow-up.

This sample size has 90% power to detect at least a 20% absolute difference in the treatment adherence rate between the intervention arm compared with the ETAU arm at 6 and/or 12 months follow-up based on two-tailed alpha of 0.025 (to account for multiple testing in respect of 6-month and 12-month testing). This is based on an observed ≈20% higher level of adherence within the intervention compared with ETAU arms in the original STOPS trial at 3 and 12 months follow-up,[11] and further assuming the design effect PHC-level parameters as noted in (2) above. Thus, the study is adequately powered to detect significant differences (at the level of two-tailed alpha of 0.05 in each case) across the two distinct primary outcomes: (1) the primary clinical outcome (GAF) and (2) the primary process outcome (medication adherence).

### Trial management and monitoring

The STOPS+ trial will be monitored in line with the protocol and the trial standard operating procedures. An independent Trial Steering Committee (TSC) which will have service user representatives will monitor trial progress. The TSC is empowered to independently review the ethical and data management procedures. A Data and Safety Monitoring Board (DSMB) will be convened to ensure the safety of participants and the integrity of the data. The PHCs in the trial will receive training around safety reporting and additionally, participants and their family members will be informed of how to report concerns and safety issues, while they are participating in the trial. The Project Management Committee will be responsible for setting up the trial, ongoing management and monitoring, promotion of the study, training and the interpretation of the results.

### Safety and adverse events reporting

All adverse events and serious adverse events (SAEs) that are reported by the participant, family member or identified by the trial team or PHC staff members during the trial will be recorded and reported following trial safety reporting procedures. Related and unexpected SAEs will be reported to the local Research Ethics Committee, DSMB and TSC within 15 days. The DSMB will review all

safety data and make any decisions regarding early termination of the study in line with their respective terms of reference.

## Data management

All data will be managed in line with the protocol and will be stored securely following the KMU and Keele University standard policy on data storage and confidentiality. Individual participant data will be pseudonymised through the use of a unique study ID and will be stored securely and separately from any identifiable data. Access to data will be restricted to members of the research team. Outcome data will be entered into an electronic database which will be stored on an encrypted secure network at KMU requiring a password to access. Data entry will be quality checked and coded using standard processes.

## Statistical analysis

Statistical analysis will be performed blind to cluster allocation. Primary analysis will be on an intention-to-treat (ITT) basis, with all participants analysed in their cluster. Between-group differences for both primary outcome measures (GAF and medication adherence) will be analysed using longitudinal mixed models that accommodate clustering at the (upper/hierarchical) practice level as well as repeated outcome measurements across three time points (baseline, 6 and 12 months) per participant, that is, linear/logistic mixed models for numerical and categorical outcomes, respectively, including PHC and participant as random factors. These models will be used to quantify the absolute between group difference in the primary clinical outcome (GAF) and odds ratio (and extracted absolute difference) for the primary process outcome (% adherence)—using linear and logistic models, respectively. For the GAF, the primary endpoint is the 'average' GAF over 6 and 12 months using available and modelled data to estimate the combined summary average mean difference in GAF between treatment arms over the 6-month and 12-month timeline; statistical significance in this case being given against two-tailed p=0.05. Though the 'average' is the single primary endpoint for the GAF, we will also evaluate the treatment-by-time interaction to give estimates of between-arm mean differences at the individual 6 and 12 months follow-up time points (secondary endpoints).

For medication adherence, between-group comparison at both 6 and 12 months is primary (derived through treatment-by-time interaction within the generalised mixed model)—hence, the requirement to take into account the multiple testing by assessing significance against a more conservative two-tailed p=0.025. Estimates of between-group mean differences will be given including 95% CIs and p values. The mixed models (detailed above) will be adjusted for age, gender, baseline GAF score and corresponding baseline score of the outcome being measured as individual-patient-level covariates, PHC-level characteristics (PHC size), stratification variable (rural/urban) and a random effect for the PHC. Mixed-model analysis fulfils the ITT principle with missing data being accounted for under the missing at random assumption.

## Cost-effectiveness

The cost of STOPS+ will be evaluated against ETAU using the Service Receipt Inventory,[28] which has been used in previous trials in Pakistan.[21] This will include the cost of primary and outpatient care visits, inpatient admissions, drug regimens, diagnostic tests, travel time, transport costs and lost days from work. Costs will be subsequently linked to outcomes in order to establish whether STOPS+ represents a cost-effective use of resources (eg, whether the additional cost of supervision by the relative are offset by the reduced hospital care and/or improvements in the clinical outcome—GAF and Quality Adjusted Life Years (QALY). Cost-effectiveness planes and acceptability curves will evaluate the uncertainty in estimates as well as the probability of STOPS+ being cost-effective across a range of willingness-to-pay thresholds. Adjusted analysis will be carried to accommodate clustering of PHCs.

## ETHICS AND DISSEMINATION

Ethical approval to conduct this trial has been obtained from Keele University Ethical Review Panel (reference: MH-190017) and Khyber Medical University Ethical Review Board (ref: DIR-KMU-EB/ST/000648). Subsequently, recruitment began on 1 November 2019. Results will be disseminated through peer-reviewed journals and academic conferences. The trial is being conducted in accordance with the International Conference on Harmonisation/Good Clinical Practice Guidelines (1996) which are consistent with principles of Declaration of Helsinki (1996) Ethical Principles for Medical Research Involving Human Subjects.

## Process evaluation

Patients and relatives, primary care physicians and MPT from participating PHCs in both arms will be invited to take part in a semistructured interview to assess the acceptability of intervention and its implementation in the health system.[29] These interviews will last approximately 30–40 min and will be audio recorded with permission. A minimum of 20 individual semistructured interviews[30] will be conducted (continuing until data saturation is reached)[31 32] with patients and their relatives to explore (a) the acceptability of the STOPS +intervention, (b) the experience/burden of the training and support for the patient's family member and (c) barriers and facilitators to medication adherence and overall satisfaction with the intervention and trial participation. A minimum of 10 interviews will be conducted with PHC physicians. In addition, we will conduct interviews with at least five decision-makers with responsibilities for developing or implementing health policy in the directorate of health in order to capture their perceptions about the feasibility, the benefits, challenges and acceptability of STOPS+

in the health system. Participants for these interviews will be purposively sampled to stratify by gender and urban/rural setting. A semistructured topic guide will be used to facilitate interviews. Informed consent will be obtained prior to the start of the interview and will include consent for the use of direct pseudonymised quotes.

An inductive, exploratory framework will be adopted using thematic analysis based on the principles of grounded theory.[33 34] A sample of early transcripts will be independently coded by the research team and a coding framework agreed on. This framework will then be applied to subsequent coding. Coded data will be analysed independently by qualitative study leads at Keele and KMU to develop categories and themes to be discussed in wider research team meetings. Constant comparison will be used to explore connections within and across transcripts and codes, highlighting consistencies and variation.[35] Analysis will follow an iterative process, with emergent findings used to further refine topic guides for subsequent interviews.

## DISCUSSION

People suffering from schizophrenia represent one of the most disadvantaged groups in LMICs facing a wide mental health treatment gap. A number of psychosocial interventions have been evaluated in RCTs and there is consensus that the treatment of schizophrenia should combine antipsychotic medication and psychosocial interventions.[36] However, there is limited evidence how these interventions can be implemented in the real-world settings.

The proposed study adopts an implementation approach addressing systemic barriers to reduce treatment gap for schizophrenia in the community. The STOPS+ intervention aims to address three major implementation barriers (1) improving supply of antipsychotic medication at primary care level, (2) involving primary care workers in the treatment of schizophrenia and (3) improving treatment adherence by involving family members in supervising treatment. The STOPS+ approach therefore goes beyond most psychosocial interventions evaluated in the treatment of schizophrenia in LMICs[37 38] in addressing the health system, community and service users-related factors to improve access to treatment.

A unique aspect of the trial is that it is powered for both process and clinical outcomes. These two primary outcomes are (1) a clinical outcome measure—GAF and (2) a process outcome measure—adherence to medication. This combined with a process evaluation carried out after completion of the cluster RCT will provide much needed evidence on the implementation of community-based interventions for schizophrenia in LMICs settings. The WHO Mental Health Action Plan (2013–2020) requires that there is an absolute increase of 20% in service coverage for severe mental disorders.[38] The results of the trial may help to understand the processes

and implementation strategies to achieve that goal and could provide an effective public health intervention for schizophrenia.[39]

A major strength of the STOPS intervention is that it is based on an infectious disease model that has been successfully used in improving treatment adherence for TB, that is, DOTS (http://www.searo.who.int/tb/topics/what_dots/en/) as a public health intervention. The DOTS model is currently being used in most LMICs, including the settings of this trial. This can potentially help in integration of the care for a non-communicable chronic disorder like schizophrenia in the public health systems of LMICs.

**Author affiliations**
[1]School of Primary, Community and Social Care, Keele University, Keele, UK
[2]Institute of Public Health & Social Sciences, Khyber Medical University, Peshawar, Khyber Pakhtunkhwa, Pakistan
[3]Medical Teaching Institution, Lady Reading Hospital, Peshawar, Khyber Pakhtunkhwa, Pakistan
[4]School of Medicine, Keele University, Keele, UK
[5]Child Mental Health Unit, University of Liverpool, Liverpool, UK
[6]Research and Innovation, Midlands Partnership Foundation Trust, Staffordshire, UK

**Acknowledgements** KD is an NIHR senior investigator. The views expressed in this paper are those of the authors and not necessarily those of the NHS, NIHR or the Department of Health and Social Care.

**Contributors** TAS wrote the first draft. All authors contributed to the conceptualisation and the design of the study. SF obtained the funding for the study, conceived and developed the intervention and study design, and led creation of the team. MFK, ZU-H, MU-H, AA, NRA and US secured the study sites and contributed to design and development of intervention, community engagement and qualitative studies leading to modification of intervention. LD contributed to design and led qualitative studies for intervention development and process evaluation. US, TAS, ZU-H and MER contributed to writing ethics applications and obtaining the ethics approvals. ML contributed significantly to the study design, analysis plan and economic evaluation part of the protocol. AR, LD, KD, CM provided critical review and contributed to the design of the study. All authors read and revised the initial manuscript and approved the final version.

**Funding** This work is supported by the Medical Research Council UK, as part of their Global Alliance for Chronic Disease programme grant number MR/S00243X/1.

**Competing interests** SF in the past 3 years has received honoraria and speaker's fees from Otsuka, Lundbeck and Sunovian pharma and also research funding from Sunovian Pharmaceutical. MU-H and MFK received funding from different pharmaceuticals to attend scientific meetings.

**Patient and public involvement** Patients and/or the public were involved in the design, or conduct, or reporting, or dissemination plans of this research. Refer to the Methods section for further details.

**Patient consent for publication** Not required.

**Provenance and peer review** Not commissioned; externally peer reviewed.

**ORCID iD**
Thomas Andrew Shepherd http://orcid.org/0000-0002-8311-7452

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
