## [Reviewer comments · BMJ Open]

ARTICLE DETAILS

TITLE (PROVISIONAL)	Supervised Treatment in Outpatients for Schizophrenia Plus (STOPS+): Protocol for a cluster randomised trial of a community-based intervention to improve treatment adherence and reduce the treatment gap for schizophrenia in Pakistan
AUTHORS	Shepherd, Thomas; Ul-Haq, Zia; Ul-Haq, Mian; Khan, Muhammad Firaz; Afridi, Adil; Dikomitis, Lisa; Robinson, Michelle; Lewis, Martyn; Rahman, Atif; Dziedzic, Krysia; Saeed, Umaima; Awan, Naila; Mallen, Christian; Farooq, Saeed

VERSION 1 – REVIEW

REVIEWER	Pavla Cermakova National Institute of Mental Health, Klecany, Czech Republic
REVIEW RETURNED	12-Dec-2019

GENERAL COMMENTS	This is a well written protocol for an important study. I have a few minor suggestions:  - please add a date when the study will be performed - please address other reasons for the treatment gap in the introduction, such as gender, disability, rural setting, somatic health. For example, Kagstrom et al suggest that rural setting is associated with treatment gap for mental disorders in the Czech Republic (Eur Psychiatry. 2019 Jun;59:37-43. doi: 10.1016/j.eurpsy.2019.04.003. Epub 2019 Apr 19. The treatment gap for mental disorders and associated factors in the Czech Republic Kagstrom A et al.) As the urban vs. rural setting is an important issue in your study, it would be good to problematize it. - Please specify the ICD 10 codes according to which you will include the patients into the study.
---

REVIEWER	Samuel Law University of Toronto, Toronto, Canada
REVIEW RETURNED	24-Jan-2020

GENERAL COMMENTS	Abstract:  1. Primary outcome using GAF seems to be open further to one of the major limitations of the study as GAF is well known to be subjective and has large inter-rater reliability problems. 2. Introduction: 3. Would be helpful to define what treatment gap is, as it has a more accepted official definition in public health and epidemiology. Some comparisons to other LMIC and more developed settings world wide may be useful.
---

4. Describing or summarizing the findings of the STOP would be helpful. Otherwise we seemed to be asked to take the writers' word for it to be worthwhile project to be further studied.
5. Similarly, DOTS could use brief description on it 's merits and locally useful rationale.

Methods.

1. Several spelling and grammatical errors. Please review. (e.g. Page 7, top paragraph, two grammatical errors: "outpatient" department? And access not "assess". STOP+ paragraph, etc.).
2. Page 8. First mention of traditional healers. Please elaborate more how this may be incorporated and relevance. (I do think it is a good idea). In fact, mentioning the culturally-bespoke intervention Is first seen here in Methods and should be part of the strong reasons argued in the Introduction to do this project in Pakistan.
3. Inclusion criteria would need to include the requirement of a family member who can monitor.
4. Primary outcome measure GAF- This instrument is quite vulnerable to biases/ subjectivity introduced by the assessors. The inter-rater reliability is 50-80%, well known to be so so an instrument. Also multiple number of research assistants in this study is going to further compound this. While it is acknowledged that it is practical, it is not rigorous. DO the authors have other instrument in mind? Something more objective, and concrete- visits to hospital/ emergency, PHC, to other healers, or even PANSS, BPRS, etc?
5. Also a little description on the adherence scale/instrument would be good instead or referring to another published document for the general reader/reviewer.
6. "During all assessments, the primary outcome measures (treatment adherence scale and GAF) will be completed first to minimise the risk of bias in the event of unmasking and, if it occurs, the point of unmasking will be recorded" – while this is a good step to avoid biases, it is inherently difficult to do as GAF requires a comprehensive knowledge of the patient that is only feasible after interviewing the patient in some detail. Consider this another weakness of the chosen primary outcome instrument.
7. Blinding process - This section is well thought out- I wonder as well one more layer should be considered: as the intervention is quite family intensive, and communication on this approach is likely to occur in the community through various media and kinship channels, and information quarantine" is not likely simply by different

	PHC physical location alone. More measures need to be done to ensure non-contamination at this level. 8. Similar step would be appropriate for the clinicians at PHC 9. Page 10- data collection- this project aims to utilize family effectively- one also wonders how the project will prepare the patient for this seemingly “distrustful”, potential aspect of the good intention. Previously independent patients’ reaction need to be addressed.
--	---

VERSION 1 – AUTHOR RESPONSE

Reviewer: 1

Comment 1. This is a well written protocol for an important study.

We are grateful for the encouraging comments by the reviewer.

**Comment 2. I have a few minor suggestions:
- please add a date when the study will be performed.**

Recruitment began on the 1st of November 2019. This has been added to the paper in the Method section in the manuscript text on page 12.

Comment 3. please address other reasons for the treatment gap in the introduction, such as gender, disability, rural setting, somatic health. For example, Kagstrom et al suggest that rural setting is associated with treatment gap for mental disorders in the Czech Republic (Eur Psychiatry. 2019 Jun;59:37-43. doi: 10.1016/j.eurpsy.2019.04.003. Epub 2019 Apr 19. The treatment gap for mental disorders and associated factors in the Czech Republic Kagstrom A et al.) As the urban vs. rural setting is an important issue in your study, it would be good to problematize it.

A section has been added to the first paragraph of the Introduction section on page 4. Factors related to the treatment gap, particularly in rural settings is addressed in the paragraph.

Comment 4. Please specify the ICD 10 codes according to which you will include the patients into the study.

These have been added to the eligibility criteria on page 8.

Reviewer: 2

Comment 1. Primary outcome using GAF seems to be open further to one of the major limitations of the study as GAF is well known to be subjective and has large inter-rater reliability problems.

We recognise that whilst the GAF is sometimes criticised for its subjective nature. This is probably true for most instruments used in psychiatric research. GAF is one of the most frequently used outcome measure in mental health research ((see for example Amri et al, 2014)). The GAF was used in our previous paper (Farooq et al , 2011) and had acceptable psychometric properties.

It is important to use GAF in the STOPS+ trial will allow us to continue to build links not just to our previous intervention development work, but to other similar projects and literature. We have thoroughly trained the research assistants in this and previous research to minimise the risk of subjectivity. It is important to note that other measures are also to be used in our assessment battery including the Brief Psychiatric Rating scale (BPRS). Other measures were considered by the research team, such as the WHO-DAS, however this suffers from the same subjective limitations and will not have the advantage of linking to the previous work done to develop this project and the STOPS+ intervention. It is likely that this challenge will also form part of the discussion section in the write up of the trial results.

Comment 2. Introduction:

Would be helpful to define what treatment gap is, as it has a more accepted official definition in public health and epidemiology. Some comparisons to other LMIC and more developed settings world wide may be useful.

A treatment gap definition has been added to the Introduction on page 4 and more information relating to factors associated with the treatment gap has been added.

Comment 3. Describing or summarizing the findings of the STOP would be helpful. Otherwise we seemed to be asked to take the writers' word for it to be worthwhile project to be further studied.

More detail has been added to the description of the findings from the original STOPS study. This information has been added to the Introduction section on page 4.

Comment 4. Similarly, DOTS could use brief description on it 's merits and locally useful rationale.

Extra detail has been added into the Introduction section on page 4.

Comment 5. Several spelling and grammatical errors. Please review. (e.g. Page 7, top paragraph, two grammatical errors: “outpatient” department? And access not “assess”. STOP+ paragraph, etc.).

These have now been corrected throughout.

Comment 6. Page 8. First mention of traditional healers. Please elaborate more how this may be incorporated and relevance. (I do think it is a good idea). In fact, mentioning the culturally-bespoke intervention is first seen here in Methods and should be part of the strong reasons argued in the Introduction to do this project in Pakistan.

Traditional healers were consulted as part of the Patient Public Involvement and Engagement (PPIE) and work that was done as part of the STOPS+ intervention development. The intervention and running of the trial were both acceptable to local traditional healers (and other stakeholder groups – patients with schizophrenia, family members, health professionals and policy makers). Traditional healer’s views on the STOPS+ and how we refined the intervention from the previous STOPS iteration were of particularly high value. Whilst traditional healers have no current formal role in the operationalisation of the STOPS+ intervention they will play a large role in the further patient and public involvement (PPIE) work relating to dissemination and implementation.

Comment 7. Inclusion criteria would need to include the requirement of a family member who can monitor.

This has been added to the eligibility criteria on page 8.

Comment 8. Primary outcome measure GAF- This instrument is quite vulnerable to biases/ subjectivity introduced by the assessors. The inter-rater reliability is 50-80%, well known to be so so an instrument. Also multiple number of research assistants in this study is going to further compound this. While it is acknowledged that it is practical, it is not rigorous. DO the authors have other instrument in mind? Something more objective, and concrete- visits to hospital/ emergency, PHC, to other healers, or even PANSS, BPRS, etc?

Please see response to Reviewer 2 comment 1

It is important to note that GAF is only one of the measure being used, mainly to assess the functioning. BPRS , as suggested by the reviewer is included as one of the measure.

Comment 9. Also a little description on the adherence scale/instrument would be good instead or referring to another published document for the general reader/reviewer.

This has now been added to page 7. This item was used in our previous work exploring adherence. Using the same item will help us to build a link to that previous work and other published literature.

Comment 10. “During all assessments, the primary outcome measures (treatment adherence scale and GAF) will be completed first to minimise the risk of bias in the event of unmasking and, if it occurs, the point of unmasking will be recorded” – while this is a good step to avoid biases, it is inherently difficult to do as GAF requires a comprehensive knowledge of the patient that is only feasible after interviewing the patient in some detail. Consider this another weakness of the chosen primary outcome instrument.

We thank the reviewer for noting this positive element of the procedure that we have developed. We agree that assessment for GAF will require knowledge of the other aspects of patients' life. As indicated in the response to the above point, the research assistants will undergo comprehensive training in the administration of this instrument to minimise the bias.

Comment 11 & 12. Blinding process - This section is well thought out- I wonder as well one more layer should be considered: as the intervention is quite family intensive, and communication on this approach is likely to occur in the community through various media and kinship channels, and information quarantine” is not likely simply by different PHC physical location alone. More measures need to be done to ensure non-contamination at this level. & point 8. Similar step would be appropriate for the clinicians at PHC

In our previous study (Farooq et al, 2011) we used the individual randomisation and contamination was a major limitation. In this study the cluster design is being used to overcome this risk, as this offers one the best strategy to overcome the risk of contamination. In any cluster randomised controlled trial there will be a risk of contamination between sites. We have taken all possible measures to minimise the contamination, as described in the protocol. The Primary Healthcare Centre (PHCs) that are the units of randomisation are in different areas. This will help to minimise this risk of contamination. However, we do acknowledge this risk and it will be a limitation of the study.

Comment 13. Page 10- data collection- this project aims to utilize family effectively- one also wonders how the project will prepare the patient for this seemingly “distrustful”, potential aspect of the good intention. Previously independent patients’ reaction need to be addressed.

We are not sure if we have fully understood the reviewer's comment here, however if they are referring to the importance of the relationship between the family member and the schizophrenia patient, this is something that we agree is crucial to the operationalisation of this intervention and of course the trial. The qualitative development work (paper in preparation) that preceded the translation of the original STOPS into the STOPS+ community-based intervention model, evidenced the

acceptability of the intervention and process to the family member observer and the patient. Additionally, the training that will be given to all participants and family member observers will promote this relationship. There was also no report of such issues in our previous STOPS study that also used family member observers.

REFERENCES

Farooq S, Nazar Z, Irfan M, et al. Schizophrenia medication adherence in a resource-poor setting: randomised controlled trial of supervised treatment in out-patients for schizophrenia (STOPS). *Br J Psychiatry* 2011;**199**(6):467-472.

Amri I, Millier A, Toumi M. Minimum Clinically Important Difference in the Global Assessment Functioning in Patients with Schizophrenia. *Value in Health* 2014;**17**:A765–A766.

VERSION 2 – REVIEW

REVIEWER	Pavla Cermakova National Institute of Mental Health - Czech Republic
REVIEW RETURNED	25-Mar-2020

GENERAL COMMENTS	My comments have been addressed.
----------------------------------

REVIEWER	Samuel Law University of Toronto, Canada
REVIEW RETURNED	27-Apr-2020

GENERAL COMMENTS	It has been a pleasure to see the revised paper. You have addressed the concerns I raised well, and I would recommend full acceptance at this point. Not to delay the publishing process, I should mention there are still minor grammatical issues in the paper. E.g. words such as "months' ", sentence on "power" and placements of period before or after the [ref] could use checking throughout, etc. Also to clarify your question on my comments: "the distrust" I raised is the potential problem between the patient and the family member who is now constantly (for good intention) checking and reminding the patient, possibly provoking some reaction of being intruded upon, and feeling distrusted. This is central to a methodology like this one and could be more proactively addressed in the methodology and in the training activities for the family. (Having a qualitative study is good but being proactive is better). Also, the cultural appropriateness of involving the local and traditional healers is important enough to warrant mentioning in the introduction, where you mention the family is receptive and willing. That was what I meant by "mentioning the first time" in my comments. I am highly supportive of your acknowledging the importance of traditional healers in a study setting where the treatment gap for schizophrenia is SO high, and people by any
---

	imagination are likely using traditional healers. (involving them in the qualitative study would be very interesting). Thank you and best wishes.
--	---

VERSION 2 – AUTHOR RESPONSE

Reviewer 1:

No further changes.

We thank Reviewer 1 for their original comments. We believe that the amendments that we have made have strengthened our paper.

Reviewer 2:

It has been a pleasure to see the revised paper. You have addressed the concerns I raised well, and I would recommend full acceptance at this point. Not to delay the publishing process, I should mention there are still minor grammatical issues in the paper. E.g. words such as "months' ", sentence on "power" and placements of period before or after the [ref] could use checking throughout, etc.

We thank Reviewer 2 for noting these comments. These amendments have now been made and have been marked by track changes.

Also, to clarify your question on my comments: "the distrust" I raised is the potential problem between the patient and the family member who is now constantly (for good intention) checking and reminding the patient, possibly provoking some reaction of being intruded upon, and feeling distrusted. This is central to a methodology like this one and could be more proactively addressed in the methodology and in the training activities for the family. (Having a qualitative study is good but being proactive is better).

We thank Reviewer 2 for the clarity given on this point. We agree with the reviewer that some distrust is likely when patient medication is supervised by a relative. This may potentially become coercive. We would like to point out that it is a normal cultural practice in the setting for the study for a family member to be regularly involved in the treatment of a patient. We are aware of these concerns and training in STOPS+ is devised to address these issues. We address these concerns during the training for Medical Technicians (MTs) who will be training the relatives to supervise the medication. This will help to minimise the intrusive nature of supervision. We have added the fact that there is training with the family on this topic (as indicated by the track changes). This element of the training was also present in our previous STOPS study.

Also, the cultural appropriateness of involving the local and traditional healers is important enough to warrant mentioning in the introduction, where you mention the family is receptive and willing. That was what I meant by "mentioning the first time" in my comments. I am highly supportive of your acknowledging the importance of traditional healers in a study setting where the treatment gap for schizophrenia is SO high, and people by any imagination are likely using traditional healers. (involving them in the qualitative study would be very interesting).

We agree with this point completely and have added the traditional healers to the Introduction section as suggested.

Editorial requests

Please reformat the abstract so that it follows the structured abstract recommended in the journal's instructions for authors for study protocols.

See: https://bmjopen.bmj.com/pages/authors/#study_protocols. The Methods and Analysis sections should be combined into one section.

This has now been amended in line with the author instructions.

Along with your revised manuscript, please provide an English language examples of the patient consent form as a supplementary file as per item #32 of the SPIRIT checklist.

The patient consent form has been included in the submission as a supplementary material.

Please provide the trial registration number at the end of your Abstract.

The ISRCTN is included at the of the Abstract.

Please clarify whether the participants will given written or oral consent to participate.

The following text is included in the section 'Participant Identification' "Those interested in taking part will be asked to give written informed consent. For participants who cannot read and write, witnessed oral consent and a thumb print in lieu of a signature will be used."

Please reformat the main text so that it follows the structure recommended in the journal's instructions for authors for study protocols, for example the Ethics and Dissemination section should be a main section in the manuscript rather than a subsection.

See: <https://bmjopen.bmj.com/pages/authors/#protocol>

These changes have now been implanted.